# Reaching underserved South Africans with integrated chronic disease screening and mobile HIV counselling and testing: A retrospective, longitudinal study conducted in Cape Town

**Philip John Smith**[1]*, **Dvora Joseph Davey**[1,2,3], **Hunter Green**[2], **Morna Cornell**[4], **Linda-Gail Bekker**[1]

**1** Faculty of Health Sciences, The Desmond Tutu HIV Centre, Institute for Infectious Disease and Molecular Medicine, University of Cape Town, Cape Town, South Africa, **2** Department of Epidemiology, Fielding School of Public Health, University of California, Los Angeles, California, United States of America, **3** Faculty of Health Sciences, Division of Epidemiology and Biostatistics, School of Public Health and Family Medicine, University of Cape Town, Cape Town, South Africa, **4** Faculty of Health Sciences, Centre for Infectious Disease Epidemiology and Research, School of Public Health and Family Medicine, University of Cape Town, Cape Town, South Africa

* Philip.Smith@hiv-research.org.za

**Data Availability Statement:** Deidentified data are available for access at 10.25375/uct.14034653.v1.

## Abstract

### Background

Community-based, mobile HIV counselling and testing (HCT) and screening for non-communicable diseases (NCDs) may improve early diagnosis and referral for care in underserved populations. We evaluated HCT/NCD data and described population characteristics of those visiting a mobile clinic in high HIV disease burden settings in Cape Town, South Africa, between 2008 and 2016.

### Methods

Trained counsellors registered patients ≥12 years old at a mobile clinic, which offered HCT and blood pressure, diabetes (glucose testing) and obesity (body mass index) screening. A nurse referred patients who required HIV treatment or NCD care. Using multivariable logistic regression, we estimated correlates of new HIV diagnoses adjusting for gender, age and year.

### Results

Overall, 43,938 individuals (50% male; 29% <25 years; median age = 31 years) tested for HIV at the mobile clinic, where 27% of patients (66% of males, 34% of females) reported being debut HIV testers. Males not previously tested for HIV had higher rates of HIV positivity (11%) than females (7%). Over half (55%, n = 1,343) of those previously diagnosed HIV-positive had not initiated ART. More than one-quarter (26%) of patients screened positive for hypertension (males 28%, females 24%, p<0.001). Females were more likely overweight

**Funding:** The Metropolitan Health Group and Abbott Laboratories funded the mobile clinic. The funder had no role in study design, data collection and analysis, decision to publish, or preparation of the manuscript.

**Competing interests:** The Metropolitan Health Group and Abbott Laboratories funded the mobile clinic. This does not alter our adherence to PLOS ONE policies on sharing data and materials.

(25% vs 20%) or obese (43% vs 9%) and presented with more diabetes symptoms than males (8% vs 4%). Females (3%) reported more symptoms of STIs than males (1%). Reporting symptoms of sexually transmitted infections (aOR = 3.45, 95% CI = 2.84, 4.20), diabetes symptoms (aOR = 1.61, 95% 1.35, 1.92), and TB symptoms (aOR = 4.40, 95% CI = 3.85, 5.01) were associated with higher odds of a new HIV diagnosis after adjusting for covariates.

## Conclusion

Findings demonstrate that mobile clinics providing integrated HCT and NCD screening may offer the opportunity of early diagnosis and referral for care for those who delay screening, including men living with HIV not previously tested.

## Introduction

Although access to and uptake of HIV testing in South Africa have improved, nearly half of the adult population who were recently diagnosed with HIV were unaware of their seropositivity [1], impeding efforts to bring the South African HIV epidemic under control. Antiretroviral treatment (ART) can reduce HIV related illness and mortality and an ART-based prevention-treatment toolkit can reverse the global HIV epidemic [2,3]. A successful treatment cascade provides early and regular testing opportunities, linkage to HIV care for those who test HIV-positive, and retention to achieve undetectable viral loads [4–7].

South Africa also has high levels of untreated diabetes [8] and hypertension [9,10], and tuberculosis (TB) related mortality [11]. Only one fifth of diabetes [8] cases and less than a tenth of hypertension [10] cases were controlled on treatment. Community-based, active case-finding services may be strategic complementary platforms for earlier detection of diabetes and hypertension, before health crises occur [12]. The South African National Health and Nutrition Survey (SANHANES) diagnosed diabetes in 9.5% of the study population, and 11.2% in the Western Cape [13]. The same survey diagnosed prehypertension and hypertension at 10.4% and 10.2% in the study, and 13.4% and 9.4% in the Western Cape. Diagnostics for these conditions are efficiently conducted in community-based settings, outside of clinics and the diagnostics are more likely to be taken up by people who may delay or avoid attending a clinic until symptoms of illness prompt a clinic visit [14]. It is therefore reasonable to decentralise screening for HIV, TB, and common NCDs to improve uptake and subsequent treatment initiation for those at risk of late presentation at diagnostic services [14–17]. Differentiated platforms that integrate HCT and chronic disease screening may have confluent benefit through offering convenient pathways for underserved populations to enter the HIV treatment cascade [18] and NCD screening [12,19].

Many South Africans choose not to visit conventional clinics because multiple real and perceived barriers prevent access. These barriers include a fragmented healthcare system, congested clinics, unfriendly staff, restricted clinic work hours, multiple visits, and stigma and privacy concerns [20–23]. The effect of these barriers appears to be more pronounced among males and young people, who have poorer rates of HIV testing than females and older adults [24], delaying the start of lifesaving ART [6,25–30] and viral suppression [31]. These challenges have led to South Africa's well-established community-based HCT programs through home testing, mobile clinic testing, workplace testing and self-testing [32]. However, HCT is still done predominantly in health facilities [33–35]. Despite the move towards community-based

HIV counselling and testing (HCT), many South Africans, especially men and young people, remain unaware of their HIV status until they present at a clinic or health facility for services related to pregnancy, HIV-related illness, or a non-communicable disease (NCD)[9,21,36–38].

Scalable, differentiated models are required to provide HIV testing to communities with a high HIV disease burden and suboptimal levels of HIV and NCD screening, treatment, and control. Mobile clinic services are cost-effective [39] and have consistently been popular amongst populations who have lower levels of HCT at conventional facilities, including males and young people. Mobile services have the potential to provide rapid ART initiation [40] for underserved populations [32,41–43]. Additionally, mobile diagnostic units can offer integrated services, which provide a convenient "one-stop" point-of-care screening and referral clinic [14]. Mobile clinics that integrate NCD screening have the added benefit of reducing stigma associated with HIV testing [32,44]. While the project was not designed as a research study, understanding mobile clinic usage may inform improvements to HIV testing, treatment and prevention services to those who may delay or avoid conventional services.

## Methods

### Design

We analysed routinely collected retrospective data from a mobile HCT/NCD clinic over a seven-year period, from March 2008 to August 2016. The mobile clinic service did not operate during 2013 due to a funding shortage.

### Setting

During the study period, the mobile clinic operated five days a week in peri-urban and under-served locations in the Cape Town metropolitan area. The mobile clinic visited locations with high pedestrian traffic such as shopping malls and commuter transport hubs. These locations were in high HIV disease burden communities chosen in partnership with the Western Cape Department of Health. In an attempt to normalize HCT, the mobile clinic provided free patient-initiated HCT in combination with screening for chronic diseases (e.g. diabetes, hypertension, and obesity) and other communicable diseases (e.g. tuberculosis and sexually transmitted infections [STI]). Family planning counselling and contraception services were offered beginning in 2014. The mobile clinic was operated by a driver, two nurses, three counsellors, and an educator, and consisted of a modified van with a private consultation room, three counselling rooms, and a lavatory. The lead nurse was trained as clinical nursing practitioner, a specialised category of nurse able to provide health assessment and patient care management. The counsellors and the educator were trained in HIV counselling and testing. All staff were trained in Good Clinical Practice [45] and Human Subjects Protection [46]. The nurse provided regular, informal training for the counsellors on a range of sexual and reproductive health issues. A tent pitched outside the mobile clinic provided additional space for waiting and group counselling.

In the absence of formal studies, patients visited the mobile clinic as a routine 'wellness service' on their own initiative. A fingerprint-based biometric system was used to log and identify patients and record medical information. Between August 2008 and August 2010, some male patients were incentivized to test for HIV at the mobile clinic [47]. These patients were recruited with the help of a partner organization called Men at the Side of the Road (MSR). A recruiter employed by MSR invited unemployed men registered with MSR to attend the mobile clinic on a predetermined day and venue. Men in the incentive group were compensated with an R80 food voucher (~9.6 USD), while men in the non-incentivised group did not receive a voucher. The incentive study recruited 3,723 men in the incentive condition and

4,985 men in the standard of care. The HIV yield in the incentive condition was 16.5% (n = 617) and 5.5% (n = 276) in the standard of care condition (p < 0.001). After 2010, the mobile clinic did not formally advertise or actively recruit patients. At the time there were mobilisation campaigns ongoing in Cape Town, the Western Cape Province, and South Africa [48].

## Measurements

After verbal consent, patients were asked if they had tested for HIV before and rapid HIV testing was conducted based on the criteria of the Western Cape [16]. HIV testing included serial testing with a first line test and a second confirmatory test. Over the study period, the clinic used different HIV tests supplied by Western Cape Department of Health over the years. The most recent test used was Collodial Gold (first line test), and Abon HIV tests, Abbott Diagnostics (confirmatory tests). Patients newly diagnosed HIV positive underwent CD4 count testing onsite using a PIMA Analyser (Alere, Waltham, MA) point-of-care machine (CD4 count tests were also completed upon request for patients who were previously diagnosed HIV positive). These newly diagnosed HIV-positive patients were then issued a referral letter to their local or preferred health care facility. Clinics were not notified of individual case referrals. Referrals were based on current ART eligibility criteria. Before 2012, those with a CD4 count <200, 200–350 and >350 cells/µl were directed to link to care in one, three, and six months respectively [49]. Between 2012 and 2016, those with a CD4 count of ≤500 cells/µl were referred to initiate ART at the nearest public clinic, and those with a CD4 count of >500 cells/µl were directed to HIV care at the same clinic, based on the South African ART Guidelines [50,51]. In 2016, the South African guidelines followed the World Health Organization (WHO) recommendation for universal test and treat (UTT) [52,53]. These guidelines formed the basis for stratifying the CD4 count in the analysis. The data in this analysis were captured before UTT was implemented. Due to some patients leaving before being offered a CD4 count test, not all newly diagnosed patients received a result. Referral completion was documented and published in two prior studies [49,54].

All patients were screened for STIs by answering questions about the presence of symptoms, including genital discharge, genital sores, and pain when urinating for males, and genital discharge, genital sores, pain when urinating, and pain during sexual intercourse for females. The clinic nurse referred patients who had STI symptoms to their nearest clinic for further assessment and care. All patients were also screened for TB symptoms using a questionnaire.

Patients were screened for diabetes, body mass index (BMI), and elevated blood pressure. These NCDs are part the Department of Health's focus areas for improved diagnostics [55]. While other NCDs were not the focus of the mobile clinic, the nurse addressed other health related concerns upon patients' request. Patients were screened for diabetes symptoms (frequent urination, unexplained weight loss or gain, increased thirst, and unexplained fatigue) and a finger-stick point-of-care glucose test (a random glucose >11 and a fasting glucose >7 were considered above normal). BMI was classified as underweight (<18.5), normal (≥18.5 –<25), overweight (≥25 –<30), and obese (≥30). According to guidelines published by the South African Hypertension Society, blood pressure was categorised into the highest level of the following: normal (<120 systolic and <80 diastolic), optimal (120–129 or 80–84 diastolic), high normal (130–139 systolic or 85–89 diastolic), grade 1 hypertension (140–159 systolic or 90–99 diastolic), grade 2 hypertension (160–179 systolic or 100–109 diastolic), grade 3 hypertension (≥180 systolic or ≥110 diastolic), and isolated systolic (≥140 systolic and <90 diastolic) [56]. Each clinic room was fitted with two cuff sizes to ensure the appropriate size was used to measure blood pressure. If blood pressure was elevated, patients were seated for at least

five minutes before a second reading was taken. Patients with elevated readings were encouraged to screen again within one month. Patients were referred for monitoring at their preferred health provider if their random blood glucose was ≥11.1 mmol/L, blood pressure was systolic ≥160 mmHg and/or diastolic ≥ 100 mmHg at two time points during the visit.

Experienced counsellors called newly HIV diagnosed individuals seven days after HIV diagnosis to assess if they had attended a local clinic for HIV care or were on ART and to provide additional counselling as needed. Pregnancy tests were conducted for newly-diagnosed females.

## Data analysis

Between 2008 and 2011 the clinic team recorded patient data on paper clinic forms which were double captured on Microsoft Access. From 2013 until 2016 patient data was captured on password protected electronic tablet devices and data were uploaded to a password protected cloud database. Data quality and completion were checked weekly by the site manager and compiled for monthly reporting to the Western Cape Department of Health.

Individuals <12 years of age were excluded from the analysis. When an individual presented multiple times to the mobile clinic during the study period, only the first visit was included in this analysis. We categorised the data into patients who had ever tested for HIV versus those who were first time testers to show the characteristics of patients by prior HIV testing. Categorical variables were reported as frequencies and percentages. Continuous variables were presented as medians and interquartile ranges. $\chi^2$ and t-tests were used for comparisons between groups. To understand covariates associated with new HIV diagnoses in clients accessing community testing, we ran multivariate logistic regression models. The outcome was HIV serostatus after testing (HIV negative vs. newly diagnosed HIV positive). We excluded those with known HIV positive results from the analysis. We adjusted for a priori confounders, gender, age and year of testing. Each model was run independently and we present crude odds ratios and adjusted odds ratios and 95% confidence intervals. Missing variables were excluded from the models and presented in Table 3.

In multivariable analyses, we included variables that were potential confounders. The purpose of the regression analysis was to understand correlates of new HIV positive diagnoses. This outcome helps to understand the yield of HIV positive diagnoses by populations characteristics and by year over between 2008 and 2016. Statistical analyses were performed using STATA version 16 (StataCorp, 2019).

## Ethics

The University of Cape Town's Research Ethics Committee (REC) and Partners Human Subjects Institutional Review Board approved data acquisition. Participants gave verbal informed consent for HCT, which was recorded by the counsellor. While a parent was required to provide consent for paediatric populations under 12 years, these data are not reported here. These data were stored on password-protected computers and only clinic staff had access to these records. This research analysed routinely collected programme data.

## Results

### Demographic and clinical characteristics

Between 2008 and 2016, 43,938 individuals (50% male) visited the mobile clinic (Table 1). The median age was 32 years (IQR = 25–42) in men and 30 years (IQR = 22–41) in women. On average, $\bar{x}$ = 37 (interquartile range [IQR] 28–43) patients visited the mobile clinic per day.

**Table 1. Demographic characteristics of patients at the mobile clinic 2008–2016 (n = 43,938) by prior HIV testing.**

| | Total | | First time HIV testing | | Previously tested for HIV | |
|---|---|---|---|---|---|---|
| | n | % (column) | n | % (column) | n | % (column) |
| **Total** | 43938 | | 11770 | | 32016 | |
| **Age (median 31 years, males 32, females 30)** | | | | | | |
| 12–14 | 587 | 1% | 475 | 4% | 112 | 1% |
| 15–16 | 1242 | 3% | 635 | 5% | 598 | 2% |
| 17–19 | 3535 | 8% | 1184 | 10% | 2341 | 7% |
| 20–24 | 7297 | 17% | 1653 | 14% | 5611 | 18% |
| 25–34 | 13379 | 30% | 2524 | 21% | 10818 | 34% |
| 35–44 | 8928 | 20% | 1986 | 17% | 6920 | 22% |
| 45+ | 8960 | 20% | 3310 | 28% | 5610 | 18% |
| **Sex (3 unknown)** | | % (column) | | % (row) | | % (row) |
| Male | 21757 | 50% | 7741 | 36% | 13939 | 64% |
| Female | 22178 | 50% | 4027 | 18% | 18076 | 82% |
| **Year** | | | | | | |
| 2008 | 4789 | 11% | 2126 | 44% | 2570 | 54% |
| 2009 | 7451 | 17% | 3028 | 41% | 4400 | 59% |
| 2010 | 7942 | 18% | 2751 | 35% | 5188 | 65% |
| 2011 | 5720 | 13% | 1377 | 24% | 4342 | 76% |
| 2012 | 4977 | 11% | 1139 | 23% | 3831 | 77% |
| 2014 | 4978 | 11% | 673 | 14% | 4301 | 86% |
| 2015 | 5089 | 12% | 470 | 9% | 4604 | 90% |
| 2016 | 2992 | 7% | 206 | 7% | 2780 | 93% |

More than one-quarter (29%) of patients were under the age of 25 years (24% male, 34% female) and 30% were aged 25–34 years old (Table 1).

Nearly all patients (97%) tested for HIV while visiting the mobile clinic. Of 11,770 individuals who had not previously been tested for HIV, twice as many men (n = 7741) as women (n = 4027) did not know their status (66% males, 34% females). The proportion of debut patients reporting that this was their first HIV test declined from 44% in 2008 to 7% in 2016 (Table 1).

HIV positivity was 12% (male 11%, female 14%), with 2,743 (6%) new HIV positive diagnoses and 2,462 (6%) previously diagnosed and retested at the mobile clinic (Table 2). The HIV prevalence among those receiving their debut test was 11% for males and 7% for females. New HIV diagnoses peaked in males 35–44 years (20%) and females 25–34 years (23%). For those who had previously tested, HIV prevalence was 10% for males and 15% for females. Three-quarters (n = 2072, 76%) of those diagnosed HIV positive at the mobile clinic, and more than one-third (n = 983, 40%) of those previously diagnosed HIV positive received a CD4 count. A quarter of males (n = 394) and one-fifth of females (n = 284) had a CD4 count below 250 cells/μL. Of those previously diagnosed HIV-positive (n = 2,462), over half (n = 1,343, 55%) had not yet initiated ART, and 14 (n = 6 male) had defaulted treatment.

Over 63% of patients screened positive for at least one NCD. More than one quarter (26%) of patients visiting the mobile clinic screened positive for hypertension (males 28%, females 24%, p<0.001). Males were more likely underweight than females (5% vs. 3%), while females were more overweight (25% vs 20%) or obese (43% vs 9%) than males. Females presented with higher rates of diabetes symptoms than males (8% vs 4%). For diabetes, 16.6% of those who screened positive were previously undiagnosed, and for hypertension, 60% were previously undiagnosed (not shown in the table).

**Table 2. Clinical characteristics of patients at the mobile clinic 2008–2016 (n = 43938).**

| | Total | | First time HIV testers | | Previously tested for HIV | |
|---|---|---|---|---|---|---|
| | n | % (column) | n | % (column) | n | % (column) |
| **HIV result** | | | | | | |
| Negative | 37532 | 85% | 10428 | 89% | 26967 | 84% |
| Known Positive | 2462 | 6% | 0 | 0% | 2459 | 8% |
| New Positive | 2743 | 6% | 1091 | 9% | 1645 | 5% |
| Not Tested/Missing | 1201 | 3% | 251 | 2% | 945 | 3% |
| **HIV prevalence by sex and age** | | (prevalence) | | (prevalence) | | (prevalence) |
| Male | 2262 | 11% | 830 | 11% | 1428 | 10% |
| 12–14 | 2 | 1% | 0 | 0% | 2 | 7% |
| 15–16 | 2 | 1% | 1 | 1% | 1 | 1% |
| 17–19 | 18 | 1% | 6 | 1% | 12 | 2% |
| 20–24 | 115 | 4% | 62 | 5% | 53 | 3% |
| 25–34 | 860 | 12% | 318 | 15% | 540 | 11% |
| 35–44 | 866 | 18% | 306 | 20% | 558 | 17% |
| 45+ | 399 | 9% | 137 | 8% | 262 | 10% |
| Female | 2943 | 14% | 261 | 7% | 2676 | 15% |
| 12–14 | 1 | 1% | 0 | 0% | 1 | 1% |
| 15–16 | 21 | 3% | 6 | 2% | 15 | 3% |
| 17–19 | 109 | 5% | 17 | 3% | 92 | 6% |
| 20–24 | 409 | 11% | 38 | 9% | 371 | 11% |
| 25–34 | 1330 | 22% | 78 | 23% | 1249 | 22% |
| 35–44 | 728 | 19% | 65 | 15% | 661 | 19% |
| 45+ | 344 | 8% | 57 | 4% | 286 | 11% |
| **CD4 count by sex** | | % (column) | | % (column) | | % (column) |
| Male | | | | | | |
| <50 | 20 | 1% | 4 | 1% | 16 | 2% |
| 50–249 | 374 | 25% | 139 | 23% | 234 | 26% |
| 250–349 | 296 | 20% | 127 | 21% | 168 | 19% |
| 350–499 | 316 | 21% | 113 | 18% | 203 | 23% |
| > = 500 | 501 | 33% | 233 | 38% | 266 | 30% |
| Female | | | | | | |
| <50 | 13 | 1% | 1 | 1% | 12 | 1% |
| 50–249 | 271 | 18% | 40 | 20% | 231 | 17% |
| 250–349 | 280 | 18% | 37 | 18% | 243 | 18% |
| 350–499 | 357 | 23% | 43 | 21% | 314 | 23% |
| > = 500 | 627 | 41% | 83 | 41% | 541 | 40% |
| **HIV Treatment**[**] | | | | | | |
| Male | | | | | | |
| Not on ART | 455 | 61% | | | 455 | 61% |
| Previously on ART—defaulted | 6 | 1% | | | 6 | 1% |
| On ART | 279 | 38% | | | 279 | 38% |
| Treatment unknown | 3 | 1% | | | 3 | 1% |
| Female | | | | | | |
| Not on ART | 888 | 52% | | | 888 | 52% |
| Previously on ART—defaulted | 8 | 1% | | | 8 | 1% |
| On ART | 821 | 48% | | | 818 | 48% |
| Treatment unknown | 2 | 1% | | | 2 | 1% |

(*Continued*)

**Table 2.** (Continued)

| | Total | | First time HIV testers | | Previously tested for HIV | |
|---|---|---|---|---|---|---|
| | n | % (column) | n | % (column) | n | % (column) |
| **Blood Pressure (p<0.001)** | | | | | | |
| Male | | | | | | |
| Normal (<120/<80) | 5192 | 24% | 1839 | 24% | 3327 | 24% |
| Optimal (120-129/80-84) | 4447 | 21% | 1585 | 21% | 2841 | 21% |
| High Normal (130-139/85-89) | 5920 | 27% | 2132 | 28% | 3778 | 27% |
| Grade 1 Hypertension (140-159/90-99) | 2679 | 12% | 810 | 11% | 1859 | 13% |
| Grade 2 Hypertension (160-179/100-109) | 1144 | 5% | 437 | 6% | 706 | 5% |
| Grade 3 Hypertension (≥180/≥110) | 763 | 4% | 300 | 4% | 461 | 3% |
| Isolated Systolic (≥140/<90) | 1441 | 7% | 596 | 8% | 841 | 6% |
| Female | | | | | | |
| Normal | 7206 | 33% | 1175 | 29% | 6001 | 34% |
| Optimal | 4740 | 22% | 781 | 20% | 3940 | 22% |
| High Normal | 4668 | 21% | 835 | 21% | 3820 | 21% |
| Grade 1 Hypertension | 2563 | 12% | 456 | 11% | 2101 | 12% |
| Grade 2 Hypertension | 1047 | 5% | 248 | 6% | 799 | 4% |
| Grade 3 Hypertension | 752 | 3% | 203 | 5% | 548 | 3% |
| Isolated Systolic | 971 | 4% | 287 | 7% | 680 | 4% |
| **BMI (p<0.001)** | | | | | | |
| Male | | | | | | |
| Underweight | 999 | 5% | 394 | 5% | 599 | 4% |
| Normal | 14204 | 66% | 5473 | 71% | 8690 | 63% |
| Overweight | 4383 | 20% | 1326 | 17% | 3042 | 22% |
| Obese | 1958 | 9% | 504 | 7% | 1447 | 11% |
| Female | | | | | | |
| Underweight | 628 | 3% | 180 | 5% | 447 | 2% |
| Normal | 6511 | 30% | 1410 | 35% | 5078 | 28% |
| Overweight | 5501 | 25% | 937 | 23% | 4546 | 25% |
| Obese | 9348 | 43% | 1473 | 37% | 7850 | 44% |
| | | (prevalence) | | (prevalence) | | (prevalence) |
| **STI symptoms (any) (p<0.001)** | 1034 | 2% | 128 | 1% | 904 | 3% |
| Male | 298 | 1% | 60 | 1% | 238 | 2% |
| Female | 736 | 3% | 68 | 2% | 666 | 4% |
| **TB symptoms (any) (p = 0.993)** | 2403 | 5% | 488 | 4% | 1910 | 6% |
| Male | 1192 | 5% | 318 | 4% | 872 | 6% |
| Female | 1211 | 5% | 170 | 4% | 1038 | 6% |
| **Diabetes symptoms (any) (p<0.001)** | 2492 | 6% | 300 | 3% | 2189 | 7% |
| Male | 802 | 4% | 145 | 2% | 655 | 5% |
| Female | 1690 | 8% | 155 | 4% | 1534 | 9% |

** includes patients who have previously tested.

## Correlates of new positive HIV diagnosis

Male sex was associated with an increased risk of being newly diagnosed with HIV in mobile HIV services in the crude analysis (OR = 1.19, 95% CI = 1.10, 1.29), but this association did not persist after adjusting for age and year (Table 3). In the multivariable model, being between age 35–44 compared with 12–18 years (aOR = 5.24, 95% CI = 4.13, 6.65), reporting STI

**Table 3. Logistic regression models of associations with new HIV-positive diagnosis in mobile testing van in Cape Town (n = 40,277).**

| Characteristics | Total | | Newly diagnosed HIV-positive | | HIV-negative | | Crude OR | Adjusted OR* |
|---|---|---|---|---|---|---|---|---|
| | n | % (column) | n | % (row) | n | % (row) | | |
| **Total** | 40275 | 100 | 2743 | 7 | 37532 | 93 | | |
| **Gender** (n = 3 missing) | | | | | | | | |
| Female | 19593 | 49 | 1224 | 6 | 18369 | 94 | Reference | |
| Male | 20679 | 51 | 1519 | 7 | 19160 | 93 | **1.19 (1.10–1.29)** | 0.98 (0.91–1.06) |
| **Age** (n = 10 missing) | | | | | | | | |
| 12–18 | 3904 | 10 | 77 | 2 | 3827 | 98 | Reference | |
| 19–24 | 8288 | 21 | 376 | 5 | 7912 | 95 | **2.36 (1.84–3.03)** | **2.29 (1.79–2.94)** |
| 25–34 | 12053 | 30 | 1116 | 9 | 10937 | 91 | **5.07 (4.01–6.41)** | **4.81 (3.81–6.09)** |
| 35–44 | 7914 | 20 | 801 | 10 | 7113 | 90 | **5.60 (4.42–7.10)** | **5.24 (4.13–6.65)** |
| 45+ | 8106 | 20 | 372 | 5 | 7734 | 95 | **2.39 (1.86–3.06)** | **2.24 (1.75–2.87)** |
| **STI symptoms (any)** | 873 | 2 | 137 | 16 | 736 | 84 | **2.63 (2.18–3.17)** | **3.45 (2.84–4.20)** |
| **TB symptoms (any)** | 1939 | 5 | 360 | 19 | 1579 | 81 | **3.44 (3.05–3.89)** | **4.40 (3.85–5.01)** |
| **Diabetes symptoms (any)** | 2101 | 5 | 163 | 8 | 1938 | 92 | 1.16 (0.98–1.37) | **1.61 (1.35–1.92)** |
| **Year** | | | | | | | | |
| 2008 | 4585 | 11 | 418 | 9 | 4167 | 91 | **1.96 (1.61–2.39)** | **1.98 (1.62–2.42)** |
| 2009 | 7050 | 18 | 592 | 8 | 6458 | 92 | **1.79 (1.48–2.17)** | **1.85 (1.52–2.24)** |
| 2010 | 7211 | 18 | 635 | 9 | 6576 | 91 | **1.89 (1.56–2.28)** | **1.90 (1.57–2.30)** |
| 2011 | 5046 | 13 | 327 | 6 | 4719 | 94 | **1.35 (1.10–1.66)** | **1.44 (1.17–1.77)** |
| 2012 | 4411 | 11 | 164 | 4 | 4247 | 96 | **0.76 (0.60–0.95)** | 0.82 (0.65–1.03) |
| 2014 | 4410 | 11 | 204 | 5 | 4206 | 95 | 0.95 (0.76–1.18) | 1.01 (0.81–1.26) |
| 2015 | 4705 | 12 | 264 | 6 | 4441 | 94 | 1.16 (0.94–1.43) | 1.22 (0.99–1.51) |
| 2016 | 2857 | 7 | 139 | 5 | 2718 | 95 | Reference | |

* Adjusted model included: Gender, age, and year.

symptoms (aOR = 3.45, 95% CI = 2.84, 4.20), reporting diabetes symptoms (aOR = 1.61, 95% CI = 1.35,1.92) and reporting TB symptoms (aOR = 4.40, 95% CI = 3.85, 5.01) were associated with higher odds of a new HIV positive diagnosis. Visiting the mobile clinic in 2008 compared with 2016 (aOR = 1.98, 95% CI = 1.62, 2.42) was associated with higher odds of being newly diagnosed with HIV after controlling for other variables.

## Discussion

In this study we found that over seven years, the mobile clinic successfully provided HIV testing to over 43,000 individuals, including more than 11,500 people who had never tested for HIV before, and successfully screened for blood pressure, diabetes, STIs, TB and obesity in those attending the clinic. It is notable that the mobile clinic was able to reach men and young people under 25 years living with HIV who were previously undiagnosed. Men represented 50% of those tested and 55% of those testing with a new HIV-positive diagnosis. Of those who had been previously diagnosed, fewer than half reported being on ART. Over 63% of patients had symptoms for at least one NCD.

Men were more likely to screen positive for hypertension, but less likely to be overweight or screen positive for diabetes. Elevated blood pressure ($\geq$140 systolic or $\geq$90 diastolic) was present in 28% of males and 24% of females visiting the mobile clinic. In the SANHANES study, 10.2% were diagnosed hypertensive nationally, and 9.4% in the Western Cape [13]. Of those presenting to the mobile clinic, a high proportion of females were overweight (25%) or obese

(43%) than their male counterparts (20% overweight, 9% obese). To contrast, a higher proportion of men (5%) were underweight than women (3%). In SANHANES, 24.8% and 39.2% of females and 20.1% and 10.6% of men were overweight and obese respectively (11), whereas men had a higher proportion of underweight (5%) than women (3%). In SANHANES, 12.8% of males and 4.2% of females were underweight. Though this study did not routinely conduct glucose or glycated Haemoglobin (HbA1c) monitoring, routine diabetes symptom screening found that males (4%) were less likely to screen positive than females (8%).

While it is striking that more than half of those previously diagnosed HIV positive (61% male, 52% female) reported they had not initiated treatment, it is encouraging that these people were willing to retest at the mobile clinic. Receiving an HIV positive diagnosis can be overwhelming and some may take time to accept the diagnosis. Those who delay treatment initiation may do so because it is difficult to accept their HIV positive status due to a range of personal and social factors, such as shock, distress, shame, denial, stigma and self-stigmatisation [57,58]. Although it is ideal to start treatment as soon after diagnosis to achieve rapid viral suppression, it is good that those who had delayed initiating treatment were willing to be retested, which is a step towards acceptance.

The integrated mobile HCT/NCD clinic successfully reached debut HIV testers, with a quarter of patients testing for the first time ever. Specifically, two thirds of debut HIV testers were male, and men were as likely to use the service as females throughout the investigation period. This is an important finding, since men are less likely than women to test for HIV across Sub-Saharan Africa [32,44]. Globally there have been attempts to improve service delivery to underserved populations, particularly those who have never tested. This platform shows promise for debut testers, demonstrating that mobile clinics can offer an entry point into healthcare services for hard-to-reach populations. Providing integrated services may be more attractive to these groups, with a range of easily accessible services. To illustrate, while the integration of HIV services was introduced partly out of the need to cope with the burden of care [59], this approach has increasing evidence which indicates that integration is feasible [60] and may improve health outcomes for TB [61] and NCDs [62]. There is broad support globally for integrating HIV services due to the benefit for patients in the form of efficiency and cost saving [63–65]. While studies have reported high acceptability, feasibility, and potential for penetration, contextual evaluations of cost, fidelity, and sustainability are required. Our study did not offer ART for patients diagnosed with HIV. However, prior studies that included treatment with screening services increased uptake of these services [62]. Though the limited evidence suggests that integrated services may lower the cost per patient compared with standalone HCT services [62,63], linkage-to-care compared with test-and-treat is less well understood [66]. Additional data is required to understand treatment initiation at mobile clinics, ongoing retention preferences and viral suppression.

Increased uptake of HCT among debut HIV testers supports previous research with men in the region, suggesting that decentralised services are better able to reach and serve men [44]. It was evident that over time there were fewer people testing for the first time. Over the seven years of the study, there was gender parity in males and females visiting the mobile clinic, in contrast with far higher uptake of HIV testing in females nationally [24].

Similarly, even though the mobile clinic did not specifically target young people, almost half of patients under 20 years reported that they had not previously tested for HIV. Our findings suggest that mobile clinics can provide an entry point for young people in underserved populations who do not ordinarily attend conventional HIV testing facilities. Mobile clinics may be ideal test-and-treat platforms for rapid ART initiation in these populations. Using mobile clinics to promote earlier ART initiation in men and young people can reduce HIV related morbidity and mortality and improve life expectancy [39].

The high proportion of young people accessing the service suggests that integrated mobile clinics could be used to prevent HIV infection in this population, and to initiate treatment for those who are already HIV positive but have not yet attended a conventional clinic. Offering HIV testing and other health services on mobile clinics has the potential for scale up of ART and Pre-Exposure Prophylaxis (PrEP), recommended for individuals who are at increased risk of HIV infection [67]. Using PrEP safely in high disease burden contexts could be supported by improved access to convenient HIV testing to monitor HIV status.

The HIV prevalence by age and sex reflects the national HIV prevalence trends in which male prevalence lagged behind females until 35 years and over. Males who were diagnosed HIV positive at the mobile clinic and those who had been previously diagnosed were more likely than females to have lower CD4 count results, suggesting that on average these males were living with HIV for a longer period than their female counterparts prior to diagnosis (The mobile clinic stopped conducting CD4 counts in 2016 based on SA National guidelines). Testing in antenatal clinics may account for more frequent testing in women.

The mobile service targeted high HIV disease burden communities based on DoH recommendations to provide services in communities which ordinarily do not access conventional clinics. Based on these gathered data, the process of location selection could be refined to ensure more strategic targeting of HIV prevalence, undiagnosed, and untreated HIV. Geospatial mapping could be used to improve the provision of sexual health services to those most at risk of HIV acquisition and onward transmission. In this way, the mobile clinic could reduce time spent in communities with lower HIV prevalence and increase time spent in communities with the highest burden of HIV and NCDs. Active case-finding with mobile clinic services to communities with the highest burden of HIV and NCDs can improve the clinic's cost-effectiveness [39]. This is an important consideration in a context of constrained resources. The HIV yield for men outside of the incentives study was 8.1%, which was lower than men who had received an incentive (16.6%), but higher than the yield in the control condition (5.5%) [47]. Offering an incentive not only increased the HIV positivity yield in men who received the incentive, but the incentive may have also had a mitigating effect on HIV yield in the control condition. Not receiving an incentive may have reduced men's motivation to test in men so that only those with high intrinsic motivation presented on days that were not incentivised [68].

While screening and treatment for hypertension, diabetes, STIs and HIV are available in community clinics, many delay visiting a healthcare facility due to various personal, economic and social barriers. NCD screening was an additional benefit of the mobile clinic, and were differently distributed by sex. Levels of elevated blood pressure were high, particularly among men. Nearly half the women were obese, and STI symptoms and diabetes symptoms were higher in women than men. Since men are less likely than women to screen for chronic diseases due to lower engagement with health services, mobile clinics may be an ideal platform for men to screen. Additionally, since there were high levels of NCDs, mobile clinics may present a community-based disease screening option for those who may otherwise delay health-seeking until illness necessitates presentation at a health facility [9]. There is potential public health benefit in providing routine screening for these chronic diseases which are associated with disability and mortality. Since mobile clinics are cost-effective [39], have high levels of acceptability [41], and units can be directed to high disease burden areas, these may serve as effective diagnostic units that can offer earlier disease detection, associated with earlier treatment and care [12,19]. Subsequent disease management requires meticulous planning to ensure logistical support and follow-up for patients who require community-based care, and referral for those who may prefer more conventional care settings [69].

The mobile clinic had high rates of hypertension and hyperglycaemia, which were similar to national rates. Evidently, there is a need to offer accessible diagnostic screening for these

conditions which are potentially debilitating and associated with early mortality. There may also be benefit in studying the impact of offering treatment at a mobile clinic and comparing the rates of diagnosis and treatment outcomes compared with conventional facilities. While patients were encouraged to lower their intake of refined carbohydrate and eat a varied diet with protein, fat, and unrefined carbohydrates, these recommendations are not easy to follow in a context where refined carbohydrates are more available and more affordable [70]. A study conducted in a neighbouring community found that the environment, specifically the availability of obesogenic foods, was crucial in promoting consumption of those foods [71]. Summarily, environmental design is important to influence behaviour and the associated health outcomes. As part of that design, community based mobile clinics can provide easy access to diagnostics and treat to ensure these services are closer to those who need it most.

Patients frequently identify convenience in accessing health services as a priority. In particular, men and adolescents have prioritised convenience in health services in general, and sexual health services specifically [23,44,72]. In this project, we increased access for men and adolescents by offering the opportunity to test near commuter hubs, shopping centres and public areas, closer to where they live, work and move. Furthermore, offering a streamlined, efficient service may have contributed to its popularity. In a previous study, providing mobile HIV services in high disease burden communities successfully identified people with undiagnosed HIV [39]. In addition, screening for common NCDs may have increased the acceptability of the HCT in the population.

There were a number of limitations in this study. First, while this mobile clinic targeted high HIV disease burden communities, uptake relied upon self-selection. This means that those who were at high risk of HIV acquisition and onward transmission but chose not to test were excluded from prevention and treatment. While the clinic could ensure confidentiality of patient medical records and privacy in the consultation room, the public nature of the clinic meant that patients' attendance was visible. However, a recent study conducted with adolescents and young adults attending a mobile service showed that most (96%) rated the service as confidential [41]. Future research could investigate methods of providing HIV services that would increase uptake for those who are even harder to reach and for those who have reasons to avoid knowing their status. Previous HIV testing history was obtained via self-report, which is potentially unreliable. Patients may want a repeat test to confirm an HIV positive status, or may want to re-initiate treatment and so seek an HIV test and a referral letter. Linkage to care and ART initiation were not recorded consistently in this data, however, we previously reported that 67% of those who tested HIV-positive linked to care and 42% had initiated treatment within three months [54].

## Conclusion

Our study suggests that integrated mobile HCT/NCD clinics may help to reach underserved populations in Cape Town. Mobile clinics complement conventional clinic services by strategically targeting underserved populations (males and those under 25 years), thereby improving the HIV-positivity yield and linking them to care. Offering NCD screening and HCT may normalize HIV testing in hard-to-reach groups. Finally, integrating rapid NCD screening into HCT services can help diagnose and refer people not in care to facilities to start on diabetes, and hypertension treatment.

## Supporting information

**S1 File. Mobile HCT table of studies conducted at the mobile clinic.**
(DOCX)

## Acknowledgments

We thank the clinic team, who provided friendly services to their patients.

## Author Contributions

**Conceptualization:** Philip John Smith, Morna Cornell, Linda-Gail Bekker.

**Data curation:** Philip John Smith.

**Formal analysis:** Philip John Smith, Dvora Joseph Davey, Hunter Green.

**Methodology:** Philip John Smith, Dvora Joseph Davey.

**Project administration:** Philip John Smith.

**Supervision:** Philip John Smith.

**Writing – original draft:** Philip John Smith.

**Writing – review & editing:** Philip John Smith, Dvora Joseph Davey, Hunter Green, Morna Cornell, Linda-Gail Bekker.

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
