## [Decision Letter · Decision Letter 0]

23 Oct 2020

PONE-D-20-27383

Reaching underserved South Africans with integrated chronic disease screening and mobile HIV counselling and testing: a retrospective, longitudinal study conducted in Cape Town

PLOS ONE

Dear Dr. Smith,

Thank you for submitting your manuscript to PLOS ONE. After careful consideration, we feel that it has merit but does not fully meet PLOS ONE’s publication criteria as it currently stands. Therefore, we invite you to submit a revised version of the manuscript that addresses the points raised during the review process.

Thanks for submitting this interesting paper. Two reviewers with lots of experience in this field have very thoroughly reviewed the paper and have a number of helpful suggestions that I would encourage you to carefully consider if you choose to submit a revised manuscript.

In addition to the points raised by the two reviewers, I would ask that you address the following points regarding the regression modelling and the methods used to measure and define hypertension:

- It would be helpful to be much clearer what the purpose of the logistic regression analysis was here - it's not clearly articulated and that makes it difficult to judge whether the methods are appropriate

- I would suggest not using the term 'predictive' or 'predictors' in this context as it's not clear that you have used a predictive modelling framework

- In the Methods section, there should be a bit more detail about how you built your model. You should also provide specific information about you handled missing data

- I agree with the comments from reviewer #2 about the importance of providing more info about how blood pressure was measured and how hypertension was defined, whether this was consistent with national and international guidance etc. If not then this should be mentioned as an important limitation and as likely to be contributing to the very high prevalence of hypertension in your sample. Also, your definitions of hypertension stages don’t seem to be consistent with South African or international guidelines. Please check and clarify what guidance your definitions were based upon.

We look forward to receiving your revised manuscript.

Kind regards,

Richard John Lessells, BSc, MBChB, MRCP, DTM&H, DipHIVMed, PhD

Academic Editor

PLOS ONE

Journal Requirements:

2. In ethics statement in the manuscript and in the online submission form, please provide additional information about the patient records used in your retrospective study. Specifically, please ensure that you have discussed whether all data were fully anonymized before you accessed them and/or whether the IRB or ethics committee waived the requirement for informed consent. If patients provided informed written consent to have data from their medical records used in research, please include this information

3. Please correct your reference to "p=0.000" to "p<0.001" or as similarly appropriate, as p values cannot equal zero.

4.We note that you have indicated that data from this study are available upon request. PLOS only allows data to be available upon request if there are legal or ethical restrictions on sharing data publicly. For information on unacceptable data access restrictions, please see http://journals.plos.org/plosone/s/data-availability#loc-unacceptable-data-access-restrictions.

Reviewers' comments:

Reviewer's Responses to Questions

**Comments to the Author**

1. Is the manuscript technically sound, and do the data support the conclusions?

Reviewer #1: Yes

Reviewer #2: Partly

2. Has the statistical analysis been performed appropriately and rigorously? 

Reviewer #1: Yes

Reviewer #2: Yes

3. Have the authors made all data underlying the findings in their manuscript fully available?

Reviewer #1: Yes

Reviewer #2: No

4. Is the manuscript presented in an intelligible fashion and written in standard English?

Reviewer #1: Yes

Reviewer #2: Yes

5. Review Comments to the Author

Reviewer #1: This is a very interesting description of programmatic findings for an integrated HIV/NCD mobile unit that prioritized men and young individuals. Overall, the manuscripts reads very well and the reader has a clear understanding of the potential of mobile units to reach men in particular. Findings for HIV and NCD are remarkable, with NCD rates including hyperglycemia and hypertension extremely high among the client population which warrants further discussion and consideration, given that HIV is largely the focus of the manuscript.

Introduction:

1. It is clear why the focus is on HIV within the given context including the overview of the successful treatment cascade, however, there is less emphasis on NCDs. The authors provide important information on low rates of diagnosis and treatment for NCDs but do not offer a comparable description of why there are barriers that reduce NCD diagnosis and treatment. Why is it important to specifically consider HIV and NCD integration? How do they interact? Why were they the focus of the program?

Methods:

1. Why did the counselor only provide telephone follow-up for newly diagnosed clients and not NCD clients?

2. Was referral completion documented?

3. Why was the focus only on diabetes and hypertension? Were other NCD’s considered (e.g., hypercholesterolemia, chronic lung disease, mental health, etc.)

4. Data collection: How was the information collected, recorded, verified, reported?

5. A description of the training required for the nurses, counselor and educator would be helpful.

6. What about consenting pediatric populations; was it possible to also do that verbally or was a parent required to provide their consent?

7. In the result it appears that a fair number of individuals who tested positive for HIV did not have a CD4 count, did they have to consent to the CD4 count separately?

8. The discussion touches on the fact that men and individuals under 25 were targeted for mobile clinic services, but there isn’t any information in the methodology about how under 25s were effectively linked to the mobile clinic.

9. I would also be curious to know if the mobile clinic went to the same locations during scheduled periods, of if different locations were accessed. Was the location selection driven by data? By planning with MSR?

10. Was any indexing conducted among individuals who tested HIV-positive?

Results:

1. How many clients on average (and range) visited the mobile unit?

2. It may not be possible within the frame of this project report/study, but it would be interesting to understand the rates of HIV and NCD and among whom at facilities within the same catchment areas to enable comparisons of mobile versus facility-based diagnosis and treatment for different populations.

3. Were the NCDs newly diagnosed, or is this data not available?

Discussion:

4. First paragraph: “The mobile clinic reached men and young people under 25 years living with HIV who were previously undiagnosed”. Consider rephrasing this. As it currently reads, one is led to understand that the mobile clinic only reached men and individuals under 25. It also says in the third paragraph that young people were not specifically targeted, so further clarification is needed.

5. Can the authors offer an explanation as to why only half of those individuals who were already diagnosed with HIV were on ART? Compare it to other literature?

6. Second paragraph: “Providing integrated services may be more attractive to these groups, with a range of easily accessible services.” Further discussion is needed here on how integrated services make access more attractive? Draw from the other literature available to share what specific aspects of integration may be increasing access, and particularly among men and young people.

7. In the third paragraph, it states that mobile platforms may be ideal for test and treat. The authors should remind the reader that they did not offer test and treat, and draw comparisons with other literature that demonstrate that test and treat in differentiated platforms, including mobile service delivery, is acceptable and feasible.

8. Overall, the discussion heavily focuses on HIV. However, the NCD data is extremely compelling and requires further consideration in and of itself. Can the authors draw conclusions as to why the NCD rates are so high? Are NCDs routinely screened at health facilities? Is treatment not readily available? What kind of follow-up counseling was provided to individuals who were overweight/obese, with hyperglycemia or hypertension?

Conclusion:

1. Last sentence, add an “and” between diabetes and hypertension.

Abstract:

1. Conclusion: From reading the abstract, the results don’t lead to the conclusion that “mobile clinics that integrate HCT and NCD offer the opportunity of early diagnosis and referral for care”. You may want to soften this language (given the aim of your study was not determining if integrated mobile services improved early access) to…”mobile clinics that provide integrated HCT and NCD may offer….”

Reviewer #2: The authors analyze retrospectively programmatic data that was collected from 2009-2016 from a mobile health unit that performed HIV counseling and testing and also screened for hypertension, diabetes, TB and STIs. They present data from a large number of participants (n=43986) making this a very valuable dataset. In the discussion the authors make a number of valuable points about the advantages of mobile clinics and their potential to reach men, young people and other hard-to-access populations. It would improve the manuscript if these points were more directly linked back to the analyses performed and if the definitions used were clarified.

It seems that over the years the mobile testing unit participated in specific research studies (vs. just the one named study?) and was also integrated into the routine Western Cape health service. This should be laid out more clearly. Table 3 shows that HIV testing positivity rates changed significantly over the years. To understand this, it is important to clarify how recruitment to the mobile unit changed over the years.

While HIV testing is clearly the focus of the manuscript, the title highlights "integrated chronic disease screening". The methods section needs improvement to indicate 1) exactly which screening tests were done for each disease and 2) how presence of each disease was defined. This information is quite unclear for blood pressure, diabetes, STIs and TB. Were symptom screens only done for STIs and TB? What questions were used? Specifically, it is not correct to classify people as "hypertensive" based on an elevated blood pressure measurement on a single day. This should be corrected throughout.

In the presentation of the results, Tables 1 & 2, use "first time HIV tester" vs. "non-first time HIV tester" as the main way of looking at the results. The reason for this is not clearly stated. Is this meant to stand as a proxy for "people who have previously not accessed traditional health care facilities" vs "people who have previously accessed traditional health care venues"? If so, please make this assumption/framework more clear in the introduction. Otherwise, consider using a different primary frame for Tables 1 & 2 (e.g. Newly-diagnosed HIV-positive as in Table 3). Disappointingly, this framework is barely used in the discussion or conclusion section. If it is to be featured so prominently in the tables, please explain the rationale for this and then discuss the results and conclusions that can be drawn from it more thoroughly.

Specific comments for improvement below:

Introduction:

1. An additional benefit of integrated HIV and NCD platform is that multi-disease screening can reduce stigma associated with HIV testing. If this is applicable in your setting, suggest adding this to the introduction or discussion.

Methods:

Design:

1. If study measurements were funded by research grants it would be appropriate to list these sources of funding more specifically than "international research funders".

Setting

2. "Family planning counseling and contraception services were beginning in 2014." - rephrase

3. Please indicate training level of nurses (Professional, Enrolled?)

4. It seems that the mobile clinic was used sometimes for research studies (funded by international funders) and sometimes as part of a DoH-funded "wellness service." Please clarify if the same measurements were conducted in both scenarios? Please also clarify how recruitment was done for each of these scenarios. The authors state that in the absence of formal studies participants arrived at the mobile clinic on their own initiative. Was community engagement conducted? Advertising? When studies were being conducted were specific groups recruited or was advertisement conducted? These details are necessary for readers to understand the sub-set of the population that was screened as this is a central point of the manuscript. A partial answer to this question is in the last paragraph of the "Setting" section. Suggest moving it up and combining it with the above quoted sentence. Was this the only study conducted using the mobile clinic over the years? Please provide a complete list of studies and their recruitment procedures (could go in supplement).

5. Add a sub-heading "Measurements" after first paragraph.

6. "Debut testing or repeat testing was obtained via self-report" - the meaning of this sentence is unclear. Does it mean that whether a given instance of HCT was debut testing or repeat testing was based on self-report? Pls clarify.

7. Which rapid HIV tests were used? Was confirmatory HIV testing performed at the mobile clinic?

8. In sentence staring "before 2012" middle CD4 range should read 200-350.

9. What is meant by "Diabetes risk factors." please define. How was blood glucose measured? Plasma, not finger stick point of care? What range was considered abnormal?

10. For blood pressure readings, were the measurements conducted in accordance with WHO-STEPS protocols including having patients rest seated for 15 minutes before the reading, using the appropriate cuff size, etc. If two readings were taken, which one was used in the analysis? Or an average? Please specify. Was a follow-up measurement conducted on a different day? If not, please be cautious about defining "hypertension" on the basis of a single day's measurement.

11. STI and TB screening are mentioned only in the context of pregnancy testing. Yet these are highlighted in the abstract as main findings. What tools were used for these screens? please clarify in a standalone sentence (as was done for diabetes) and clarify if all patients were screened for these conditions or only a subset? What action was taken as a result of these symptom screens?

Results:

1. Table 1: For the First time testing and Previously Testing columns. 1) Please indicate "HIV" testing in column headers. 2) Are column statistics the most useful here? For age it is fine, but for the other two Sex and Year row %'s would be more interpretable.

2. Page 7 last paragraph: According to the methods all HIV+ receive a point-of-care CD4 count, but here the authors give lower numbers 75% and 54%. What is the reason for this? Was it only introduced in a certain year? Pls clarify.

3. Table 2: Please indicate "HIV" testing in column headers. As in Table 1, please explain the reason for highlighting these two groups in the columns and consider use of row statistics instead of column.

4. Almost two thirds of patients visiting mobile clinic were hypertensive -- how was this defined. Should be stated clearly in the methods. Since it appears to be based on a single measurement at a single timepoint, the correct terminology is "elevated blood pressure" or you could say that they screened positive for hypertension. But it is not appropriate to conclude that people are "hypertensive" on the basis of measurement/s taken on a single day. Elevated BP on two days seperated by at least a week are required to make this diagnosis.

5. The methods state that 'plasma glucose' was measured. Why are these results not shown? It would be very useful to show these in addition to diabetes symptoms.

6. Table 3. It is quite interesting to see that HIV positive rates significantly differed based on calendar year. To assist the reader in interpreting this, please update the "settings" section of the methods to explain whether different recruitment techniques were used in different calendar years (aside from the single study that is mentioned).

Discussion:

1. Paragraph 1: "Over 69% of patients had at least one NCD" Revise phrasing - "screened positive"?. As indicated above based on a single BP measurement, hypertension cannot be diagnosed. And please explain more clearly in methods and in results section how diabetes (or a positive diabetes screen) was diagnosed.

2. Paragraph 2, sentence 1: "debut testers" - specify HIV testers.

3. The point that men used the mobile testing service is important. The fact that there was a male-focused study conducted during part of the time period reported here requires clarification. Please include a sub-analysis that shows how many of the men tested enroled in that study and received incentive to test. It is important for the reader to know whether the positive rates of male enrollment and HIV-testing reported in this manuscript are the result of the incentive study or if they were present even in the absence of specific recruitment techniques or incentive strategies.

4. Second to last sentence of Paragraph 2 requires copy-editing.

5. Paragraph 6, sentence 1. This should have been stated clearly in the "setting" section of the Methods.

6. Paragraph 6. The point made in this paragraph about the need to focus the use of the mobile clinic in areas of highest risk is well made. But the evidence supporting the conclusion stated in the second sentence is unclear. Please clarify.

7. Paragraph 7. Levels of high blood pressure were very high, but would caution over interpretation based on a single measurement at a mobile clinic. Please clarify whether participants rested in a seated position for at least 15 minutes prior to the measurement? Were appropriate cuff sizes available? Any data on follow-up measurements on another day?

8. Add to limitations: caveats (listed above) about NCD screening tests (single BP measurements), nature of diabetes screening (still unclear to me).

Funding: Please clarify. Conflicting information in different places.

6. PLOS authors have the option to publish the peer review history of their article (what does this mean?). If published, this will include your full peer review and any attached files.

Reviewer #1: **Yes: **Malia Duffy

Reviewer #2: **Yes: **Emily B. Wong

---

## [Author Response · Author response to Decision Letter 0]

15 Feb 2021

04 February 2021

Philip Smith

Desmond Tutu HIV Centre

University of Cape Town

Dear Dr Lessels,

Thank you for the review of our manuscript “Reaching underserved South Africans with integrated chronic disease screening and mobile HIV counselling and testing: a retrospective, longitudinal study conducted in Cape Town.” We have addressed the reviewers’ comments in below in bold. We hope that our manuscript finds favourable review, thank you. 

Current submission and prior presentations disclosure

This work has not been published and is not under consideration in any other peer-reviewed media.

Authorship and conflicts 

The listed authors have all contributed significantly to the design, analysis and the written work, and all authors have given final approval for the version to be published. To the best of our knowledge, no conflict of interest, financial or other, exists. 

The anonymised dataset can be located here – 10.25375/uct.14034653 . Thank you kindly for reviewing our manuscript. 

Corresponding author 

Philip Smith

Desmond Tutu HIV Centre

Institute for Infectious Disease and Molecular Medicine

Faculty of Health Sciences, University of Cape Town

Anzio Road, Observatory, Cape Town 7925, South Africa 

E: Philip.Smith@hiv-research.org.za

t: +27 (0)21 6501895 m: +27 (0)83 8702289 

Reviewer #1: This is a very interesting description of programmatic findings for an integrated HIV/NCD mobile unit that prioritized men and young individuals. Overall, the manuscripts reads very well and the reader has a clear understanding of the potential of mobile units to reach men in particular. Findings for HIV and NCD are remarkable, with NCD rates including hyperglycemia and hypertension extremely high among the client population which warrants further discussion and consideration, given that HIV is largely the focus of the manuscript.

Introduction:

1. It is clear why the focus is on HIV within the given context including the overview of the successful treatment cascade, however, there is less emphasis on NCDs. The authors provide important information on low rates of diagnosis and treatment for NCDs but do not offer a comparable description of why there are barriers that reduce NCD diagnosis and treatment. Why is it important to specifically consider HIV and NCD integration? How do they interact? Why were they the focus of the program?

Response: Thank you for reviewing our manuscript. South Africa has high levels of NCD and TB related mortality. Diagnostics for these conditions are efficiently conducted in community-based settings, outside of clinics, and the diagnostics are more likely to be taken up by people who may avoid attending a clinic until serious symptoms of illness prompt a clinic visit. It is therefore reasonable to decentralise point-of-care diagnostics to improve uptake and subsequent treatment initiation. We have updated the introduction section to clarify the inclusion of NCDs, as follows (Line 63):

“South Africa also has high levels of untreated diabetes (Stokes et al., 2017) and hypertension (Berry et al., 2017; Kane et al., 2017), and tuberculosis (TB) related mortality (Statistics South Africa, 2020). Only one fifth of diabetes (Stokes et al., 2017) cases and less than a tenth of hypertension (Berry et al., 2017) cases were controlled on treatment. Community-based, active case-finding services may be strategic complementary platforms for earlier detection of diabetes and hypertension, before health crises occur (Coleman et al., 1998). The South African National Health and Nutrition Survey (SANHANES) diagnosed diabetes in 9.5% of the study population, and 11.2% in the Western Cape (Shisana et al., 2014). The same survey diagnosed prehypertension and hypertension at 10.4% and 10.2% in the study, and 13.4% and 9.4% in the Western Cape. Diagnostics for these conditions are efficiently conducted in community-based settings, outside of clinics and the diagnostics are more likely to be taken up by people who may delay or avoid attending a clinic until symptoms of illness prompt a clinic visit (Kranzer et al., 2012). It is therefore reasonable to decentralise screening for HIV, TB, and common NCDs to improve uptake and subsequent treatment initiation for those at risk of late presentation at diagnostic services (Drain et al., 2013; Fomundam et al., 2017; Garrib et al., 2019; Kranzer et al., 2012). Differentiated platforms that integrate HCT and chronic disease screening may have confluent benefit through offering convenient pathways for underserved populations to enter the HIV treatment cascade (Maughan-Brown et al., 2019) and NCD screening (Al-Halaweh et al., 2019; Coleman et al., 1998).”

Methods:

1. Why did the counselor only provide telephone follow-up for newly diagnosed clients and not NCD clients?

Response: The mobile counsellors prioritised HIV and TB diagnoses, which are notifiable to the Department of Health in South Africa. We have updated the discussion section to discuss follow-up for NCDs (line 360).

“Additionally, since there were high levels of NCDs, mobile clinics may present a community-based disease screening option for those who may otherwise delay health-seeking until illness necessitates presentation at a health facility (Kane et al., 2017). There is potential public health benefit in providing routine screening for these chronic diseases which are associated with disability and mortality. Since mobile clinics are cost-effective (Bassett et al., 2014), have high levels of acceptability (Smith et al., 2019), and units can be directed to high disease burden areas, these may serve as effective diagnostic units that can offer earlier disease detection, associated with earlier treatment and care (Al-Halaweh et al., 2019; Coleman et al., 1998). Subsequent disease management requires meticulous planning to ensure logistical support for patients who require community-based care, and referral for those who may prefer more conventional care settings (Aye et al., 2020).”

2. Was referral completion documented?

Response: Each referred patient received up to four phone calls to document referral completion. We have previously published these data in JAIDS, AIDS patient care and STDs. We updated the methods section to reflect the documentation of clinic referrals (line 159). 

“Referral completion was documented and published in two prior studies (Govindasamy et al., 2013; Maughan-Brown et al., 2018).”.

3. Why was the focus only on diabetes and hypertension? Were other NCD’s considered (e.g., hypercholesterolemia, chronic lung disease, mental health, etc.)

Response: These NCDs are part the DoH focus areas for improved diagnostics. While other NCDs were not the focus of the mobile clinic, the nurse addressed other health related concerns upon patients’ request. These were not recorded in the medical records. 

4. Data collection: How was the information collected, recorded, verified, reported?

Response: Thank you for highlighting this gap in the paper. We have updated the manuscript to describe data collection (line 190):

“Between 2008 and 2011 the clinic team recorded patient data on paper clinic forms which were double captured on Microsoft Access. From 2013 until 2016 patient data was captured on password protected electronic tablet devices and data were uploaded to a password protected cloud database. The site manager verified data quality and completion on a weekly basis. The data were compiled and reported monthly to Western Cape Department of Health.”

5. A description of the training required for the nurses, counselor and educator would be helpful.

Response: We have described the training of the staff in the manuscript (line 119). 

“The nurse was trained as a clinic nursing practitioner, a specialised category of nurse able to provide health assessment and patient care management. The counsellors and the educator were trained in HIV counselling and testing. All staff were trained in Good Clinical Practice and Human Subjects Protection. The nurse provided regular, informal training for the counsellors on a range of sexual and reproductive health issues.”

6. What about consenting pediatric populations; was it possible to also do that verbally or was a parent required to provide their consent?

Response: We have updated the manuscript to describe consent for minors (line 208). 

“While a parent was required to provide consent for paediatric populations under 12 years, these data are not reported here.”

7. In the result it appears that a fair number of individuals who tested positive for HIV did not have a CD4 count, did they have to consent to the CD4 count separately?

Response: Patients newly diagnosed HIV positive received a CD4 count test. Due to some patients leaving before being offered a CD4 count test, not all newly diagnosed patients received a result. Patients who were previously diagnosed HIV positive occasionally requested a CD4 count test, which is why there are more CD4 count results than new HIV diagnoses. Anecdotally, these were requested for referral for initiating treatment or reinitiating treatment after defaulting (line 145). 

“Patients newly diagnosed HIV positive underwent CD4 count testing onsite using a PIMA Analyser (Alere, Waltham, MA) point-of-care machine (CD4 count tests were also completed upon request for patients who were previously diagnosed HIV positive).”

8. The discussion touches on the fact that men and individuals under 25 were targeted for mobile clinic services, but there isn’t any information in the methodology about how under 25s were effectively linked to the mobile clinic.

Response: While we did not specifically target men outside of the incentives study, or those under 25 years, it seems the nature of the mobile clinic service delivery platform improved access in this population. 

9. I would also be curious to know if the mobile clinic went to the same locations during scheduled periods, of if different locations were accessed. Was the location selection driven by data? By planning with MSR?

Response: The mobile clinic visited different locations. The locations were in part chosen in partnership with the Department of Health, and with community representatives. In this data set, locations were not chosen based on data. In the discussion we have emphasised that, in addition to partnering with the Department of Health, testing should be based on data to target communities with higher rates of undiagnosed untreated HIV. We have clarified 

10. Was any indexing conducted among individuals who tested HIV-positive?

Response: Indexing was not conducted, though counsellors asked if the patient knew his/her partner’s serostatus. 

Results:

1. How many clients on average (and range) visited the mobile unit?

Response: We have updated the manuscript to state that the mean patients per day was 37 (interquartile range [IQR] 28 – 43) (line 216). 

“On average, x̄=37 (interquartile range [IQR] 28 – 43) patients visited the mobile clinic per day.”

2. It may not be possible within the frame of this project report/study, but it would be interesting to understand the rates of HIV and NCD and among whom at facilities within the same catchment areas to enable comparisons of mobile versus facility-based diagnosis and treatment for different populations.

Response: We covered a large area in the Cape Town Metro. While there are no comparable stats for the area covered, we have added stats for the Western Cape for HIV, hypertension, and diabetes (lines 69-70). 

3. Were the NCDs newly diagnosed, or is this data not available?

Response: We have updated the manuscript to report these data (line 242). 

“For diabetes, 16.6% were previously undiagnosed, and for hypertension, 60% were previously undiagnosed.” 

Discussion:

4. First paragraph: “The mobile clinic reached men and young people under 25 years living with HIV who were previously undiagnosed”. Consider rephrasing this. As it currently reads, one is led to understand that the mobile clinic only reached men and individuals under 25. It also says in the third paragraph that young people were not specifically targeted, so further clarification is needed.

Response: Thank you for giving us the opportunity to clarify. The sentence has been rephrased (line 263): 

“It is notable that the mobile clinic was able to reach men and people under 25 years living with HIV who were previously undiagnosed, although the service did not specifically specific target groups.”

5. Can the authors offer an explanation as to why only half of those individuals who were already diagnosed with HIV were on ART? Compare it to other literature?

Response: Please see the following additional text added to the discussion about previously diagnosed patients who had not initiated ART (line 280). 

“While it is striking that more than half of those previously diagnosed HIV positive (61% male, 52% female) reported they had not initiated treatment, it is encouraging that these people were willing to retest at the mobile clinic. Receiving an HIV positive diagnosis can be overwhelming and some may take time to accept the diagnosis. Those who delay treatment initiation may do so because it is difficult to accept their HIV positive status due to a range of personal and social factors, such as shock, distress, shame, denial, stigma and self-stigmatisation (Bogart et al., 2013; Mukolo et al., 2013). Although it is ideal to start treatment as soon after diagnosis to achieve rapid viral suppression, it is good that those who had delayed initiating treatment were willing to be retested, which is a step towards acceptance.”

6. Second paragraph: “Providing integrated services may be more attractive to these groups, with a range of easily accessible services.” Further discussion is needed here on how integrated services make access more attractive? Draw from the other literature available to share what specific aspects of integration may be increasing access, and particularly among men and young people.

Response: This section has been updated with further discussion on integrated services (line 297). 

“To illustrate, while the integration of HIV services was introduced partly out of the need to cope with the burden of care (Ford et al., 2018), this approach has increasing evidence which indicates that integration is feasible (Haldane et al., 2018) and may improve health outcomes for TB (Gilbert et al., 2015) and NCDs (Kemp et al., 2018). There is broad support globally for integrating HIV services due to the benefit for patients in the form of efficiency and cost saving (Brown et al., 2019; Mabuto et al., 2014; Sweeney et al., 2012). While studies have reported high acceptability, feasibility, and potential for penetration, contextual evaluations of cost, fidelity, and sustainability are required.”

7. In the third paragraph, it states that mobile platforms may be ideal for test and treat. The authors should remind the reader that they did not offer test and treat, and draw comparisons with other literature that demonstrate that test and treat in differentiated platforms, including mobile service delivery, is acceptable and feasible.

Response: This section has included a discussion on the teat and treat literature (line 304). 

“Our study did not offer ART for patients diagnosed with HIV. However, prior studies that included treatment with screening services increased uptake of these services (Kemp et al., 2018). Though the limited evidence suggests that integrated services may lower the cost per patient compared with standalone HCT services (Kemp et al., 2018; Sweeney et al., 2012), linkage-to-care compared with test-and-treat is less well understood (Dave et al., 2019). Additional data is required to understand treatment initiation at mobile clinics, ongoing retention preferences and viral suppression.”

8. Overall, the discussion heavily focuses on HIV. However, the NCD data is extremely compelling and requires further consideration in and of itself. Can the authors draw conclusions as to why the NCD rates are so high? Are NCDs routinely screened at health facilities? Is treatment not readily available? What kind of follow-up counseling was provided to individuals who were overweight/obese, with hyperglycemia or hypertension?

Response: We have updated the methods section to describe the follow up for patients with with elevated blood pressure and those who screened positive for hyperglycaemia (line 180). 

“If blood pressure was elevated, a second reading was taken by the nurse and the lower reading was recorded. Patients with elevated readings were encouraged to screen again within one month. Patients were referred for monitoring at their preferred health provider if their random blood glucose was ≥11.1 mmol/L, blood pressure was systolic ≥160 mmHg and/or diastolic ≥ 100 mmHg at two time points during the visit since their preferred clinics could best manage these patients.”

We have updated the discussion to include elevated blood pressure and hyperglycemia (line 370). 

“The mobile clinic had high rates of hypertension and hyperglycaemia, which were similar to national rates. Evidently, there is a need to offer accessible diagnostic screening for these conditions which are potentially debilitating and associated with early mortality. There may also be benefit in studying the impact of offering treatment at a mobile clinic and comparing the rates of diagnosis and treatment outcomes compared with conventional facilities. While patients were encouraged to lower their intake of refined carbohydrate and eat a varied diet with protein, fat, and unrefined carbohydrates, these recommendations are not easy to follow in a context where refined carbohydrates are more available and more affordable (Drewnowski, 2004). A study conducted in a neighbouring community found that the environment, specifically the availability of obesogenic foods, was crucial in promoting consumption of those foods (Kroll et al., 2019). Summarily, environmental design is important to influence behaviour and the associated health outcomes. As part of that design, community based mobile clinics can provide easy access to diagnostics and treat to ensure these services are closer to those who need it most.” 

Conclusion:

1. Last sentence, add an “and” between diabetes and hypertension.

Response: We added an “and” to the last sentence. 

Abstract:

1. Conclusion: From reading the abstract, the results don’t lead to the conclusion that “mobile clinics that integrate HCT and NCD offer the opportunity of early diagnosis and referral for care”. You may want to soften this language (given the aim of your study was not determining if integrated mobile services improved early access) to…”mobile clinics that provide integrated HCT and NCD may offer….”

Response: Thank you for this useful comment. We have updated the conclusion (line 51) to state that “… that provide integrated HCT and NCD may offer…”

Reviewer #2: The authors analyze retrospectively programmatic data that was collected from 2009-2016 from a mobile health unit that performed HIV counseling and testing and also screened for hypertension, diabetes, TB and STIs. They present data from a large number of participants (n=43986) making this a very valuable dataset. In the discussion the authors make a number of valuable points about the advantages of mobile clinics and their potential to reach men, young people and other hard-to-access populations. It would improve the manuscript if these points were more directly linked back to the analyses performed and if the definitions used were clarified.

It seems that over the years the mobile testing unit participated in specific research studies (vs. just the one named study?) and was also integrated into the routine Western Cape health service. This should be laid out more clearly. Table 3 shows that HIV testing positivity rates changed significantly over the years. To understand this, it is important to clarify how recruitment to the mobile unit changed over the years.

While HIV testing is clearly the focus of the manuscript, the title highlights "integrated chronic disease screening". The methods section needs improvement to indicate 1) exactly which screening tests were done for each disease and 2) how presence of each disease was defined. This information is quite unclear for blood pressure, diabetes, STIs and TB. Were symptom screens only done for STIs and TB? What questions were used? Specifically, it is not correct to classify people as "hypertensive" based on an elevated blood pressure measurement on a single day. This should be corrected throughout.

In the presentation of the results, Tables 1 & 2, use "first time HIV tester" vs. "non-first time HIV tester" as the main way of looking at the results. The reason for this is not clearly stated. Is this meant to stand as a proxy for "people who have previously not accessed traditional health care facilities" vs "people who have previously accessed traditional health care venues"? If so, please make this assumption/framework more clear in the introduction. Otherwise, consider using a different primary frame for Tables 1 & 2 (e.g. Newly-diagnosed HIV-positive as in Table 3). Disappointingly, this framework is barely used in the discussion or conclusion section. If it is to be featured so prominently in the tables, please explain the rationale for this and then discuss the results and conclusions that can be drawn from it more thoroughly.

Specific comments for improvement below:

Introduction:

1. An additional benefit of integrated HIV and NCD platform is that multi-disease screening can reduce stigma associated with HIV testing. If this is applicable in your setting, suggest adding this to the introduction or discussion.

Response: We have updated the last paragraph of the introduction to state that mobile clinics that integrate NCD screening may reduce stigma associated with HIV testing (line 99).

“Mobile clinics that integrate NCD screening have the added benefit of reducing stigma associated with HIV testing (Sharma et al., 2015, 2017). While the project was not designed as a research study, understanding mobile clinic usage may inform improvements to HIV testing, treatment and prevention services to those who may delay or avoid conventional services.” 

Methods:

Design:

1. If study measurements were funded by research grants it would be appropriate to list these sources of funding more specifically than "international research funders".

Response: The journal has requested that, since no funding was received for the manuscript, that all funding except for commercial funding be removed. We have stated in the “Competing Interests” declaration that Metropolitan Health Group and Abbott Laboratories funded the mobile clinic. We have stated that this does not affect compliance with PLOS ONE policies.

Setting

2. "Family planning counseling and contraception services were beginning in 2014." – rephrase

Response: this sentence has been rephrased to state (line 116), “Family planning counselling and contraception services were offered from 2014.”

3. Please indicate training level of nurses (Professional, Enrolled?)

Response: This section has been updated to include the training of the nurses; professional nurses with a primary healthcare certification (line 119). 

The lead nurse was trained as a clinic nursing practitioner, a specialised category of nurse able to provide health assessment and patient care management. The counsellors and the educator were trained in HIV counselling and testing. All staff were trained in Good Clinical Practice and Human Subjects Protection. The nurse provided regular, informal training for the counsellors on a range of sexual and reproductive health issues.”

4. It seems that the mobile clinic was used sometimes for research studies (funded by international funders) and sometimes as part of a DoH-funded "wellness service." Please clarify if the same measurements were conducted in both scenarios? Please also clarify how recruitment was done for each of these scenarios. The authors state that in the absence of formal studies participants arrived at the mobile clinic on their own initiative. Was community engagement conducted? Advertising? When studies were being conducted were specific groups recruited or was advertisement conducted? These details are necessary for readers to understand the sub-set of the population that was screened as this is a central point of the manuscript. A partial answer to this question is in the last paragraph of the "Setting" section. Suggest moving it up and combining it with the above quoted sentence. Was this the only study conducted using the mobile clinic over the years? Please provide a complete list of studies and their recruitment procedures (could go in supplement).

Response: Thank you for the useful suggestions. We have moved the last paragraph of the setting to the second paragraph. We have included a list of the studies conducted at the mobile clinic during the study period. 

5. Add a sub-heading "Measurements" after first paragraph.

Response: We have added a “Measurements” sub-heading (line 139). 

6. "Debut testing or repeat testing was obtained via self-report" - the meaning of this sentence is unclear. Does it mean that whether a given instance of HCT was debut testing or repeat testing was based on self-report? Pls clarify.

Response: We have rephrased the sentence to clarify that participants were asked whether they had ever had a HIV test before or if this was a repeat test (line 140). 

“After verbal consent, patients were asked if they had tested for HIV before and rapid HIV testing was conducted based on the criteria of the Western Cape.”

7. Which rapid HIV tests were used? Was confirmatory HIV testing performed at the mobile clinic?

Response: We have added the HIV tests to the text (line 141). 

“HIV testing included serial testing with a first line test and a second confirmatory test. Over the study period, the clinic used different HIV tests supplied by Western Cape Department of Health over the years. The most recent test used was Collodial Gold (first line test), and Abon HIV tests, Abbott Diagnostics (confirmatory tests).”

8. In sentence staring "before 2012" middle CD4 range should read 200-350.

Response: We have revised the text to read 200-350 (line 151). 

9. What is meant by "Diabetes risk factors." please define. How was blood glucose measured? Plasma, not finger stick point of care? What range was considered abnormal?

Response: We have revised the text as follows (line 169): 

“Patients were screened for diabetes symptoms (frequent urination, unexplained weight loss or gain, increased thirst, and unexplained fatigue) and a finger-stick point-of-care glucose test (a random glucose >11 and a fasting glucose >7 were considered above normal).”

10. For blood pressure readings, were the measurements conducted in accordance with WHO-STEPS protocols including having patients rest seated for 15 minutes before the reading, using the appropriate cuff size, etc. If two readings were taken, which one was used in the analysis? Or an average? Please specify. Was a follow-up measurement conducted on a different day? If not, please be cautious about defining "hypertension" on the basis of a single day's measurement.

Response: Thanks for the opportunity to clarify this point. We have updated the text to elaborate (line 179).

“Each clinic room was kitted with two cuff sizes to ensure the appropriate size was used to measure blood pressure. Patients were seated for at least five minutes before the first reading. If blood pressure was elevated, a second reading was taken by the nurse, who recorded the lower reading. Patients with elevated readings were encouraged to screen again within one month. Patients were referred for monitoring at their preferred health provider if their random blood glucose was ≥11.1 mmol/L, blood pressure was systolic ≥160 mmHg and/or diastolic ≥ 100 mmHg at two time points during the visit since their preferred clinics could best manage these patients.”

11. STI and TB screening are mentioned only in the context of pregnancy testing. Yet these are highlighted in the abstract as main findings. What tools were used for these screens? please clarify in a standalone sentence (as was done for diabetes) and clarify if all patients were screened for these conditions or only a subset? What action was taken as a result of these symptom screens?

Response: The section on STI screening has been revised as follows (line 161). 

“All patients were screened for STIs by answering questions about the presence of symptoms, including genital discharge, genital sores, pain when urinating for males, and genital discharge, genital sores, pain when urinating, and pain during sexual intercourse for females. The clinic nurse referred patients who had STI symptoms to their nearest clinic for further assessment and care.”

Results:

1. Table 1: For the First time testing and Previously Testing columns. 1) Please indicate "HIV" testing in column headers. 2) Are column statistics the most useful here? For age it is fine, but for the other two Sex and Year row %'s would be more interpretable.

Response: The headers have been updated to include HIV testing. The Sex and Year columns have been updated to reflect row percentages. 

2. Page 7 last paragraph: According to the methods all HIV+ receive a point-of-care CD4 count, but here the authors give lower numbers 75% and 54%. What is the reason for this? Was it only introduced in a certain year? Pls clarify.

Response: Newly diagnosed patients received a CD4 count test. Patients who were previously diagnosed HIV positive on occasion requested a CD4 count test, which is why there are more CD4 count results than new HIV diagnoses. Anecdotally, these were requested for referral for initiating treatment or reinitiating treatment after defaulting. 

3. Table 2: Please indicate "HIV" testing in column headers. As in Table 1, please explain the reason for highlighting these two groups in the columns and consider use of row statistics instead of column.

Response: The headers have been updated to include HIV testing. We have used debut testing versus previously tested for HIV to highlight the ability of the mobile clinic to reach debut testers. This is a strength of the mobile clinic for populations such as young people and men, who frequently avoid diagnostic and treatment services (line 145).

“Patients newly diagnosed HIV positive underwent CD4 count testing onsite using a PIMA Analyser (Alere, Waltham, MA) point-of-care machine (CD4 count tests were also completed upon request for patients who were previously diagnosed HIV positive).”

4. Almost two thirds of patients visiting mobile clinic were hypertensive -- how was this defined. Should be stated clearly in the methods. Since it appears to be based on a single measurement at a single timepoint, the correct terminology is "elevated blood pressure" or you could say that they screened positive for hypertension. But it is not appropriate to conclude that people are "hypertensive" on the basis of measurement/s taken on a single day. Elevated BP on two days seperated by at least a week are required to make this diagnosis.

Response: We have updated the methods section of the manuscript to clarify the process for determining elevated blood pressure (line 166).

“Patients were screened for diabetes, body mass index (BMI), and elevated blood pressure. These NCDs are part the Department of Health’s focus areas for improved diagnostics (South African National Department of Health, 2013). While other NCDs were not the focus of the mobile clinic, the nurse addressed other health related concerns upon patients’ request. Patients were screened for diabetes symptoms (frequent urination, unexplained weight loss or gain, increased thirst, and unexplained fatigue) and a finger-stick point-of-care glucose test (a random glucose >11 and a fasting glucose >7 were considered above normal). BMI was classified as underweight (<20), normal (≥20 – <25), overweight (≥25 – <30), and obese (≥30). According to guidelines published by the South African Hypertension Society, blood pressure was categorised into the highest level of the following: normal (<120 systolic and <80 diastolic), optimal (120-129 or 80-84 diastolic), high normal (130-139 systolic or 85-89 diastolic), grade 1 hypertension (140-159 systolic or 90-99 diastolic), grade 2 hypertension (160-179 systolic or 100-109 diastolic), grade 3 hypertension (≥180 systolic or ≥110 diastolic), and isolated systolic (≥140 systolic and <90 diastolic) (Hypertension guideline working group et al., 2014). Each clinic room was fitted with two cuff sizes to ensure the appropriate size was used to measure blood pressure. If blood pressure was elevated, patients were seated for at least five minutes before a second reading was taken. Patients with elevated readings were encouraged to screen again within one month.”

5. The methods state that 'plasma glucose' was measured. Why are these results not shown? It would be very useful to show these in addition to diabetes symptoms.

Response: Thank you for pointing out this error. We have removed the reference to plasma glucose and corrected the text to state (line 169): 

“Patients were screened for diabetes symptoms (frequent urination, unexplained weight loss or gain, increased thirst and unexplained fatigue) and a finger-stick point-of-care glucose test (a random glucose >11 and a fasting glucose >7 were considered above normal).”

6. Table 3. It is quite interesting to see that HIV positive rates significantly differed based on calendar year. To assist the reader in interpreting this, please update the "settings" section of the methods to explain whether different recruitment techniques were used in different calendar years (aside from the single study that is mentioned).

Response: The settings section has been updated to describe recruitment activities (line 126):

“In the absence of formal studies, patients visited the mobile clinic as a routine ‘wellness service’ on their own initiative. A fingerprint-based biometric system was used to log and identify patients and record medical information. Between August 2008 and August 2010, some male patients were incentivized to test for HIV at the mobile clinic (Nglazi et al., 2012). These patients were recruited with the help of a partner organization called Men at the Side of the Road (MSR). A recruiter employed by MSR invited unemployed men registered with MSR to attend the mobile clinic on a predetermined day and venue. They were compensated with an R80 food voucher (~9.6 USD). After 2010, the mobile clinic did not formally advertise or actively recruit patients. A list of the studies has been included in a supplement to this manuscript.” 

Discussion:

1. Paragraph 1: "Over 69% of patients had at least one NCD" Revise phrasing - "screened positive"?. As indicated above based on a single BP measurement, hypertension cannot be diagnosed. And please explain more clearly in methods and in results section how diabetes (or a positive diabetes screen) was diagnosed. 

Response: Thanks for the chance to clarify these points. We have revised the text as follows:

(line 166), methods:

“Patients were screened for diabetes, body mass index (BMI), and elevated blood pressure. These NCDs are part the Department of Health’s focus areas for improved diagnostics (South African National Department of Health, 2013). While other NCDs were not the focus of the mobile clinic, the nurse addressed other health related concerns upon patients’ request. Patients were screened for diabetes symptoms (frequent urination, unexplained weight loss or gain, increased thirst, and unexplained fatigue) and a finger-stick point-of-care glucose test (a random glucose >11 and a fasting glucose >7 were considered above normal). BMI was classified as underweight (<20), normal (≥20 – <25), overweight (≥25 – <30), and obese (≥30). According to guidelines published by the South African Hypertension Society, blood pressure was categorised into the highest level of the following: normal (<120 systolic and <80 diastolic), optimal (120-129 or 80-84 diastolic), high normal (130-139 systolic or 85-89 diastolic), grade 1 hypertension (140-159 systolic or 90-99 diastolic), grade 2 hypertension (160-179 systolic or 100-109 diastolic), grade 3 hypertension (≥180 systolic or ≥110 diastolic), and isolated systolic (≥140 systolic and <90 diastolic) (Hypertension guideline working group et al., 2014). Each clinic room was fitted with two cuff sizes to ensure the appropriate size was used to measure blood pressure. If blood pressure was elevated, patients were seated for at least five minutes before a second reading was taken. Patients with elevated readings were encouraged to screen again within one month. Patients were referred for monitoring at their preferred health provider if their random blood glucose was ≥11.1 mmol/L, blood pressure was systolic ≥160 mmHg and/or diastolic ≥ 100 mmHg at two time points during the visit.”

(line 266), results: 

“Over 63% of patients had symptoms for at least one NCD.”

2. Paragraph 2, sentence 1: "debut testers" - specify HIV testers.

Response: We have revised the sentence as follows (line 310): 

“Increased uptake of HCT among debut HIV testers supports previous research with men in the region, suggesting that decentralised services are better able to reach and serve men.”

3. The point that men used the mobile testing service is important. The fact that there was a male-focused study conducted during part of the time period reported here requires clarification. Please include a sub-analysis that shows how many of the men tested enrolled in that study and received incentive to test. It is important for the reader to know whether the positive rates of male enrollment and HIV-testing reported in this manuscript are the result of the incentive study or if they were present even in the absence of specific recruitment techniques or incentive strategies.

Response: We have updated the manuscript to include a description of the incentivisation study (line 134). 

“During the course of the observational, non-randomised study between 2008 and 2010, 3723 men were incentivised for HIV testing and 4985 were not incentivised. This study observed an increase in HIV positive test results in men in who were incentivised compared with the non-incentivised group.” 

Additionally, we have included in the manuscript discussion (line 346): 

“The HIV yield for men outside of the incentives study was 8.1%, which was lower than men who had received an incentive (16.6%), but higher than the yield in the control condition (5.5%). Offering an incentive not only increased the HIV positivity yield in men who received the incentive, but may have also had a mitigating effect on HIV yield in the control condition. Not receiving an incentive may have reduced men’s motivation to test so that only those with high intrinsic motivation presented on days that were not incentivised”.

4. Second to last sentence of Paragraph 2 requires copy-editing.

Response: We have rephrased the sentence to state (line 312): 

“Over the seven years of the study, there was gender parity in males and females visiting the mobile clinic, in contrast with far higher uptake of HIV testing in females nationally.”

5. Paragraph 6, sentence 1. This should have been stated clearly in the "setting" section of the Methods.

Response: We have clarified in the setting section that (line 109): 

“During the study period, the mobile clinic operated five days a week in peri-urban and underserved locations in the Cape Town metropolitan area. The mobile clinic visited locations with high pedestrian traffic such as shopping malls and commuter transport hubs. These locations were in high HIV disease burden communities chosen in partnership with the Western Cape Department of Health.”

6. Paragraph 6. The point made in this paragraph about the need to focus the use of the mobile clinic in areas of highest risk is well made. But the evidence supporting the conclusion stated in the second sentence is unclear. Please clarify.

Response: We have rephrased the sentence to state that (line 338): 

“Based on these gathered data, the process of location selection could be refined to ensure more strategic targeting of HIV prevalence, undiagnosed, and untreated HIV.”

7. Paragraph 7. Levels of high blood pressure were very high, but would caution over interpretation based on a single measurement at a mobile clinic. Please clarify whether participants rested in a seated position for at least 15 minutes prior to the measurement? Were appropriate cuff sizes available? Any data on follow-up measurements on another day?

Response: We have updated the manuscript in the methods section to clarify (line 179):

“Each clinic room was kitted with two cuff sizes to ensure the appropriate size was used to measure blood pressure. Patients were seated for at least five minutes before the first reading. If blood pressure was elevated, a second reading was taken by the nurse and the lower reading was recorded. Patients with elevated readings were encouraged to screen again within one month. Patients were referred for monitoring at their preferred health provider if their random blood glucose was ≥11.1 mmol/L, blood pressure was systolic ≥160 mmHg and/or diastolic ≥ 100 mmHg at two time points during the visit since their preferred clinics could best manage these patients.”

8. Add to limitations: caveats (listed above) about NCD screening tests (single BP measurements), nature of diabetes screening (still unclear to me).

Response: We have revised the manuscript to clarify the blood pressure measurements (line 179, described above) and the diabetes screening (line 169)

“Patients were screened for diabetes symptoms (frequent urination, unexplained weight loss or gain, increased thirst, and unexplained fatigue) and a finger-stick point-of-care glucose test (a random glucose >11 and a fasting glucose >7 were considered above normal).”

Funding: Please clarify. Conflicting information in different places.

Response: The journal has requested that, since no funding was received for the manuscript, that all funding except for commercial funding for the mobile clinic be removed. We have stated in the “Competing Interests” declaration that Metropolitan Health Group and Abbott Laboratories funded the mobile clinic. We have stated that this does not affect compliance with PLOS ONE policies.

---

## [Editor Report · Decision Letter 1]

22 Mar 2021

Reaching underserved South Africans with integrated chronic disease screening and mobile HIV counselling and testing: a retrospective, longitudinal study conducted in Cape Town

PONE-D-20-27383R1

Dear Dr. Smith,

We’re pleased to inform you that your manuscript has been judged scientifically suitable for publication and will be formally accepted for publication once it meets all outstanding technical requirements.

Apologies for the delayed decision. I was waiting for comments from the reviewers but unfortunately these were not forthcoming, probably because people are busy with COVID-19 work. However, I’m happy that you have addressed the comments comprehensively - thank you for doing this. I would just ask that you make the additional changes that you communicated to me by email (that were in response to my comments) and upload that version.

Kind regards,

Richard John Lessells, BSc, MBChB, MRCP, DTM&H, DipHIVMed, PhD

Academic Editor

PLOS ONE
---

## [Editor Report · Acceptance letter]

26 Apr 2021

PONE-D-20-27383R1 

Reaching underserved South Africans with integrated chronic disease screening and mobile HIV counselling and testing: a retrospective, longitudinal study conducted in Cape Town 

Dear Dr. Smith:

I'm pleased to inform you that your manuscript has been deemed suitable for publication in PLOS ONE. Congratulations! Your manuscript is now with our production department. 

Kind regards, 

on behalf of

Dr. Richard John Lessells 

Academic Editor

PLOS ONE